# Deterministic vs. LLM-Controlled Orchestration for COBOL-to-Python Modernization

Naing Oo Lwin
Bucknell University, Lewisburg, PA, USA
Astrio, San Francisco, CA, USA
naing.oo.lwin@bucknell.edu

Rajesh Kumar
Bucknell University, Lewisburg, PA, USA
rajesh.kumar@bucknell.edu

## Abstract

Modernizing legacy COBOL systems remains difficult due to scarce expertise, large and long-lived codebases, and strict correctness requirements. Recent large language model (LLM)-based modernization systems increasingly rely on agentic workflows in which the model controls multi-step tool execution. However, it remains unclear whether delegating execution control to the LLM improves correctness, robustness, or efficiency in structured software engineering workflows. We present a controlled empirical study of deterministic and LLM-controlled orchestration for COBOL-to-Python modernization. Using a unified experimental framework, we hold the language models, prompts, tools, configurations, and source programs constant while varying only the execution control strategy. This isolates orchestration as the sole experimental variable. We evaluate both approaches using functional correctness, robustness across repeated stochastic runs, and computational efficiency. Across multiple models, deterministic orchestration achieves comparable computational accuracy to LLM-controlled orchestration while improving worst-case robustness and reducing performance variability across runs. Deterministic execution also reduces token consumption by up to 3.5×, leading to substantially lower operational cost. These results suggest that, in structured modernization workflows with explicit validation stages, fixed execution policies provide more stable and cost-efficient behavior than fully agentic orchestration without reducing translation quality.

## CCS Concepts

• **Software and its engineering** → **Software maintenance tools**;
• **Computing methodologies** → *Artificial intelligence*.

## Keywords

Legacy Code Modernization, COBOL-to-Python Translation, LLM-based Software Engineering, Agentic AI

**ACM Reference Format:**

Naing Oo Lwin and Rajesh Kumar. 2026. Deterministic vs. LLM-Controlled Orchestration for COBOL-to-Python Modernization. In *Proceedings of the 3rd ACM International Conference on AI-powered Software (AIware '26)*. ACM, New York, NY, USA, 8 pages. https://doi.org/XXXXXXX.XXXXXXX

## 1 Introduction

Legacy software systems written in COBOL remain important in domains such as finance and government, where they continue to support long-lived operational infrastructure. Maintaining and modernizing these systems has become increasingly difficult due to workforce attrition, limited learning resources, scarce documentation, and the complexity of decades-old codebases [2]. Modernization is therefore necessary not only to sustain these systems but also to integrate them with modern software ecosystems.

Recent work has explored the use of large language models (LLMs) for code translation and modernization, including systems that combine tool use, iterative reasoning, and autonomous execution [10, 15]. Many of these systems adopt agentic workflows in which the model selects tools, determines execution order, and decides when to retry or terminate execution. While such approaches have shown promise in open-ended software engineering tasks, their effectiveness in structured modernization workflows remains unclear.

Legacy modernization workflows are typically organized around explicit stages such as transformation, validation, and testing. Intermediate outputs can often be verified automatically using executable test suites and deterministic validation procedures. This raises a key question: *does delegating execution control to the LLM improve structured modernization workflows compared to fixed execution policies?* Existing work has primarily focused on generation quality or end-task performance. As a result, it remains difficult to isolate how orchestration strategy itself affects correctness, robustness, and computational cost.

In this paper, we present a controlled empirical study comparing deterministic and LLM-controlled orchestration for COBOL-to-Python modernization. Using ATLAS (Autonomous Transpilation for Legacy Application Systems), we hold the language models, prompts, tools, configurations, and source programs constant while varying only the execution control strategy. This isolates orchestration as the sole experimental variable.

In deterministic orchestration, tool ordering, validation stages, and retry behavior are fixed across runs. In contrast, LLM-controlled orchestration allows the model to dynamically determine execution flow based on intermediate outputs. This enables a direct comparison between fixed execution policies and model-driven control under identical experimental conditions.

Our results show that both approaches achieve comparable functional correctness but differ substantially in reliability and efficiency. Deterministic orchestration improves worst-case robustness and reduces variation across runs. It also reduces token consumption by up to 3.5×, leading to substantially lower operational cost.

These findings suggest that, for structured modernization workflows with explicit validation stages, deterministic execution provides a more stable and cost-efficient alternative to fully agentic orchestration without reducing translation quality.

Briefly, our contributions are as follows:

- We present a controlled experimental framework for LLM-based code modernization that isolates execution control as the sole experimental variable, enabling direct comparison between deterministic and LLM-controlled orchestration [1].
- We show that deterministic orchestration achieves comparable functional correctness to LLM-controlled orchestration while improving worst-case robustness and reducing variation across repeated runs.
- We provide a quantitative analysis of computational efficiency and show that deterministic orchestration reduces token consumption by up to 3.5×, leading to substantially lower operational cost.

The remainder of this paper is organized as follows. Section 2 reviews prior work on automated code translation and agentic LLM systems. Section 3 describes the orchestration strategies and the ATLAS framework. Section 4 presents the experimental design, dataset, and evaluation metrics. Section 5 reports the empirical results and analysis. Finally, Section 6 concludes and discusses implications for LLM-based software engineering.

## 2 Background

### 2.1 Automated Code Translation

Recent work has explored the use of large language models (LLMs) for automated code translation and modernization, framing translation as a mapping from a source program to a semantically equivalent target representation [14]. Transformer-based models and large-scale pretraining have enabled direct, single-shot translation between programming languages, including legacy languages, often without explicit parallel corpora [4, 5].

Recent studies also examine multilingual code generation and large-scale code translation. Several works analyze factors that influence translation outcomes, including prompt language and prompt design [1, 7, 8, 11]. While these approaches demonstrate promising translation quality, they primarily operate as one-pass generators and do not explicitly model execution control, iterative validation, or workflow structure. As a result, they provide limited insight into how different orchestration strategies affect correctness, robustness under repeated execution, or computational efficiency.

### 2.2 Agentic and Tool-Orchestrated LLM Systems

More recent systems introduce agentic behavior, enabling LLMs to invoke tools, perform multi-step reasoning, and iteratively refine outputs [3]. In many agentic frameworks, the LLM acts as a central orchestrator that dynamically selects tools, orders operations, and determines retry or termination behavior at runtime [9, 15]. Several works also propose orchestration and workflow engines for standardizing and evaluating agentic execution behavior [6, 16].

These designs are well-suited to open-ended or exploratory tasks, but they implicitly treat execution control as a reasoning

---

[1]Code and artifacts are available at https://github.com/astrio-ai/atlas.

task delegated to the language model. Prior work has shown that coupling planning and execution can introduce significant variability in execution behavior, where small variations in model outputs produce divergent execution paths [13]. This complicates reproducibility, stability, and cost predictability.

In contrast, structured software engineering workflows such as legacy code modernization are typically organized around fixed execution stages, explicit tool dependencies, and verifiable intermediate states [12]. However, existing agentic systems do not empirically isolate the impact of execution control strategy itself. As a result, it remains unclear whether LLM-controlled orchestration provides measurable benefits over deterministic control when other factors are held constant.

Our work addresses this gap through a controlled comparison of deterministic and LLM-controlled orchestration. We isolate execution control as the sole experimental variable.

## 3 Methodology

### 3.1 Deterministic Tool Orchestration

ATLAS is implemented as a single-agent system with deterministic tool orchestration. By deterministic orchestration, we mean that for fixed inputs, configurations, and system state, the sequence of tool invocations—including which tools are executed, their order, and the conditions under which they are triggered—remains invariant across runs. Any non-determinism in ATLAS is strictly confined to the language model's code generation and never affects execution control, tool selection, or workflow ordering.

ATLAS enforces a fixed, stage-based execution pipeline implemented within its core message-processing routine. Each run follows the same predefined sequence of stages, including code application, optional persistence, validation, and testing. Execution order is static and does not depend on language model outputs or intermediate results. Conditional branching is governed exclusively by deterministic predicates over observable system state and configuration flags, such as whether files were modified or whether specific validation stages are enabled.

Crucially, the language model does not participate in execution planning. It does not select tools, determine execution order, initiate retries, or control termination. All orchestration decisions are resolved through explicit deterministic control logic evaluated at runtime. While user interaction may influence whether execution proceeds or terminates, it does not alter the structure or ordering of the pipeline itself.

Within individual stages, ATLAS further enforces determinism through fixed strategy ordering. When multiple mechanisms are available for applying code edits or performing validation, they are attempted sequentially in a predefined order until one succeeds. This ensures that fallback behavior remains consistent and does not introduce stochastic variation.

The orchestration layer contains no random sampling, probabilistic branching, or stochastic scheduling. Execution depends solely on observable system state and configuration parameters. Consequently, identical inputs and configurations produce identical tool-call traces across runs. This contrasts with agentic systems that delegate execution control to the language model, where tool

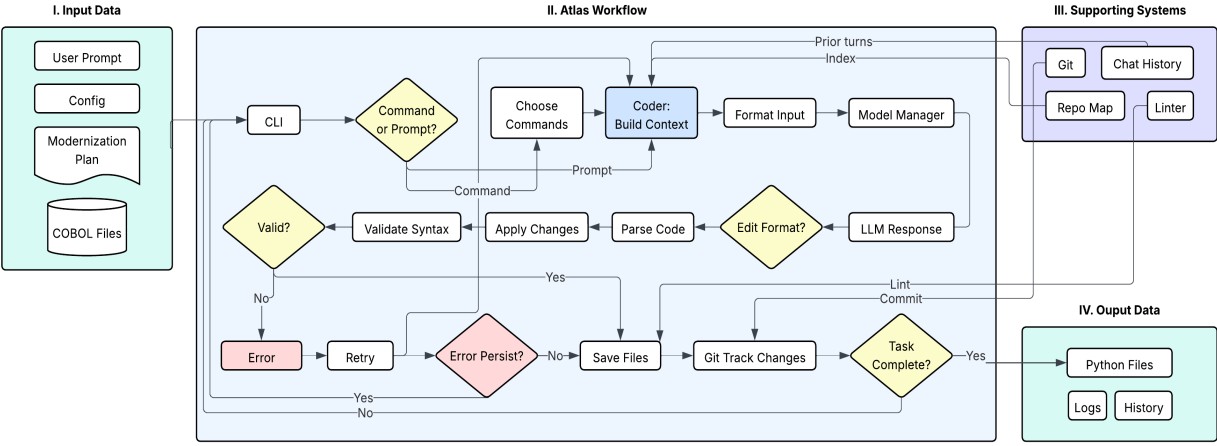

**Figure 1: Overview of the ATLAS system architecture and end-to-end COBOL-to-Python modernization workflow.**

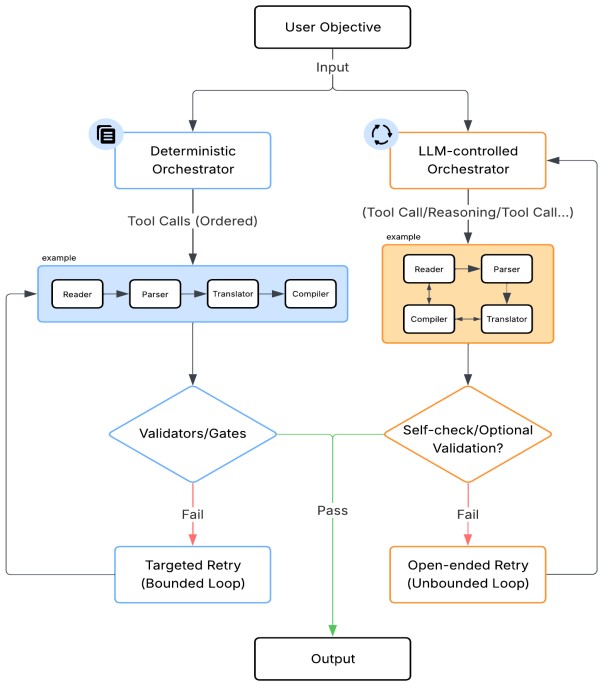

**Figure 2: Comparison of deterministic orchestration and LLM-controlled orchestration for COBOL-to-Python modernization. The deterministic lane follows a fixed execution policy with explicit tool/validator stages and bounded retries, while the LLM-controlled lane selects actions adaptively with branching and variable-length tool-use trajectories.**

selection and execution order may vary between runs, leading to increased execution variance and reduced reproducibility.

## 3.2 LLM-controlled Tool Orchestration

As a baseline, we evaluate an LLM-controlled orchestration paradigm in which the language model is responsible not only for code generation but also for execution control. In this setting, the model dynamically selects tools, determines execution order, chooses repair strategies in response to failures, and decides whether to continue or terminate execution.

In this paradigm, execution flow is not predefined. Tool invocation and ordering emerge from the language model's generation process and are influenced by intermediate outputs, prior tool results, and stochastic sampling. Consequently, identical inputs and configurations may produce different execution traces across runs, even when the same underlying language model and prompts are used.

This design reflects a common assumption in agentic coding systems that execution control can be treated as a reasoning task delegated to the language model. While this flexibility enables adaptive behavior, it also introduces variability in tool usage, execution depth, and recovery behavior. Small variations in model outputs can produce divergent sequences of operations, complicating reproducibility and execution stability.

In our experiments, this orchestration strategy serves as a contrastive baseline to deterministic execution. By holding the language model, prompts, tools, and inputs constant, we isolate the effect of LLM-controlled execution on functional correctness, robustness under repeated execution, and failure behavior in structured legacy code modernization workflows.

## 4 Experiment & Evaluation Setup

### 4.1 Experimental Objective

The objective of our experiments is to isolate the effect of execution control strategy on legacy code modernization outcomes. Specifically, we compare deterministic and LLM-controlled orchestration for COBOL-to-Python modernization while holding all other factors constant.

To enable a controlled comparison, both orchestration strategies are implemented within the same ATLAS framework and use identical language models, prompts, tools, configurations, and source programs. Execution control strategy is the sole experimental variable. This includes tool selection, execution order, retry behavior, and termination decisions.

Our evaluation focuses on three dimensions. First, we evaluate functional correctness using Computational Accuracy and Success Rate. Second, we measure robustness under repeated execution using tail-risk metrics and tool-call trace consistency. Third, we evaluate computational efficiency using tokens per successful translation. This controlled design enables direct attribution of observed differences in correctness, robustness, and operational cost to orchestration strategy rather than to model capability or prompt design.

## 4.2 Controlled Experimental Variables

To ensure a fair and controlled comparison between deterministic and LLM-controlled orchestration, we hold all experimental variables constant except for the execution control mechanism.

*Language Models.* All experiments use the same set of underlying language models. For each experiment, model architecture, version, decoding parameters (e.g., temperature), and random seed initialization protocols remain identical across orchestration strategies. This ensures that observed differences are attributable to orchestration strategy rather than to model capability or sampling configuration.

*Prompts and Instructions.* Both orchestration strategies use identical system prompts, task instructions, and formatting constraints. In both settings, the language model receives the same contextual information and code inputs without orchestration-specific prompt modifications.

*Tools.* Both orchestration strategies have access to the same tool set with identical interfaces and permissions. The available tools include:

- `read_file`: reads file contents,
- `write_file`: writes or modifies file contents,
- `list_files`: enumerates repository files,
- `web_scrape`: retrieves external web content,
- `run_command`: executes shell commands,
- `git`: executes version control operations.

No tools are added, removed, or modified between orchestration strategies.

*Inputs and Configuration.* All runs use the same source programs, test inputs, execution environment, timeout limits, and configuration flags. Validation and testing stages are enabled or disabled identically across orchestration strategies.

*Isolation of Execution Control.* Execution control is the sole experimental variable. This includes tool selection, invocation order, retry behavior, and termination decisions. By controlling all other factors, we ensure that observed differences in correctness, robustness, execution stability, and computational cost are directly attributable to orchestration strategy.

## 4.3 Dataset and Test Harness

We evaluate ATLAS using the NIST COBOL85 Test Suite, a standardized and widely used benchmark designed to validate conformance to the COBOL 85 language specification. The suite consists of several hundred self-contained COBOL programs, each paired with structured test cases that exercise specific language features and execution behaviors. Programs in our evaluation set have an average length of approximately 1,200 lines of code (LOC), representing non-trivial logic commonly found in legacy enterprise systems.

Collectively, the suite covers a broad range of COBOL constructs, including arithmetic and control flow (NC), copy and call semantics (SM, IC), intrinsic functions (IF), sorting (ST), communication (CM), debugging and reporting (DB, RW), and multiple file I/O modalities such as sequential, indexed, and relative access (SQ, IX, RL).

A key advantage of the NIST COBOL85 suite is the availability of executable test harnesses with well-defined expected outcomes. This enables automated and reproducible correctness evaluation. For each COBOL program, the suite defines a collection of test cases whose outcomes can be validated deterministically, making the benchmark well suited for evaluating functional equivalence after translation.

In our experiments, each COBOL source file is translated to Python and evaluated using the same test inputs as the original COBOL implementation. Computational Accuracy is measured by comparing the observable outputs of the translated program against those produced by the reference COBOL execution. Success Rate additionally requires that the translated program executes without runtime errors and passes all associated test cases.

By leveraging an established executable benchmark with known semantics, our evaluation avoids reliance on heuristic similarity metrics and enables direct measurement of functional correctness, robustness under repeated execution, and execution stability across orchestration strategies.

## 4.4 Evaluation Metrics

We evaluate deterministic and LLM-controlled orchestration using complementary metrics that capture functional correctness, reliability, robustness to stochasticity, and execution stability. All metrics are computed under controlled conditions while holding the language model, prompts, tools, and inputs constant.

*Computational Accuracy (CA).* Computational Accuracy measures functional equivalence between the generated program and a reference implementation over a set of test inputs:

$$\text{CA}(p) = \frac{1}{|T_p|} \sum_{t \in T_p} \mathbf{1}\big[\text{exec}(y_p, t) = \text{exec}(y_p^*, t)\big], \qquad (1)$$

where $p$ indexes a program, $T_p$ denotes its test inputs, $y_p$ is the generated program, and $y_p^*$ is the reference implementation. CA is averaged across all programs to obtain mean correctness.

*Success Rate (SR).* Success Rate measures the reliability of an orchestration strategy under repeated stochastic execution. For each source program $p$, the translation pipeline is executed $N$ independent times using different random seeds. A run is considered successful if the generated program (i) executes without runtime errors and (ii) passes all evaluation test cases.

**Table 1: Comparison of deterministic and LLM-controlled tool orchestration across models on COBOL-to-Python modernization. Metrics are averaged across programs and repeated runs. Higher is better for all metrics.**

| Model | Orchestration | CA ↑ | SR ↑ | P5-CA ↑ | $\text{CVaR}_{0.1}$ ↑ |
|---|---|---|---|---|---|
| Claude-Sonnet-4-5 | Deterministic | **0.966** | 0.902 | **0.959** | **0.953** |
| | LLM-controlled | 0.964 | **0.918** | 0.956 | 0.949 |
| GPT-5.1-Codex-Max | Deterministic | **0.969** | 0.910 | **0.962** | **0.956** |
| | LLM-controlled | 0.964 | **0.937** | 0.958 | 0.951 |
| Grok-Code-Fast-1 | Deterministic | **0.961** | 0.872 | **0.951** | **0.943** |
| | LLM-controlled | 0.958 | **0.906** | 0.941 | 0.934 |

The success rate for program $p$ is defined as:

$$\text{SR}(p) = \frac{1}{N} \sum_{i=1}^{N} \mathbf{1}[\text{run}_i(p) \text{ is successful}], \quad (2)$$

where $\mathbf{1}[\cdot]$ denotes the indicator function.

Aggregating across all programs $P$, we report the mean success rate:

$$\text{SR} = \frac{1}{|P|} \sum_{p \in P} \text{SR}(p). \quad (3)$$

*Worst-Case Robustness.* To characterize robustness under stochastic variation, we repeat each program $N$ times using independent random seeds and compute tail-risk metrics over the resulting CA values $\{CA_p^{(i)}\}_{i=1}^{N}$.

We report the 5th percentile CA:

$$\text{P5-CA}(p) = \text{Percentile}_5\left(\{CA_p^{(i)}\}\right), \quad (4)$$

which captures worst-case behavior while remaining robust to isolated outliers.

Conditional Value at Risk (CVaR) measures failure severity:

$$\text{CVaR}_\alpha(p) = \mathbb{E}\left[CA_p^{(i)} \mid CA_p^{(i)} \le P_\alpha\right], \quad (5)$$

where $\alpha = 0.1$ in our experiments. CVaR quantifies the average correctness among the worst-performing fraction of runs.

## 5 Results and Discussion

### 5.1 Main Results and Key Takeaways

Across all evaluated models and orchestration strategies, the results reveal a consistent tradeoff between correctness robustness and execution success frequency. Deterministic orchestration achieves higher computational accuracy and stronger worst-case robustness, while LLM-controlled orchestration attains higher success rates across repeated runs. Specifically, deterministic pipelines consistently achieve higher mean Computational Accuracy (CA), improved 5th-percentile accuracy (P5-CA), and higher Conditional Value at Risk (CVaR), indicating reduced tail-risk failures and more stable functional behavior.

In contrast, LLM-controlled orchestration achieves higher Success Rate (SR), indicating that adaptive agentic execution more frequently produces runnable and test-passing programs across repeated runs. Importantly, both orchestration strategies demonstrate comparable overall capability when evaluated using a single

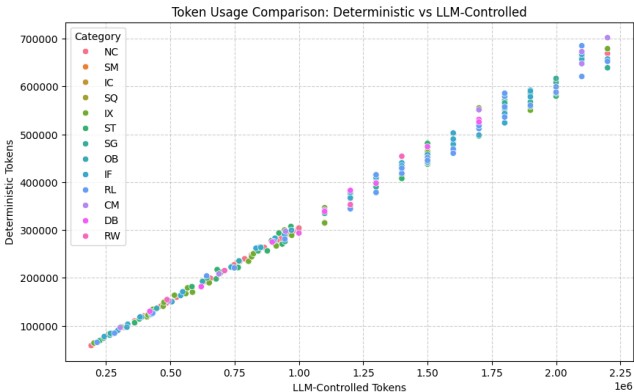

**Figure 3: Token usage per successful translation for deterministic vs. LLM-controlled orchestration.**

finalized translation per program. This suggests that orchestration strategy primarily affects execution reliability rather than underlying translation capability.

### 5.2 Correctness and Robustness Analysis

Table 1 reports aggregated performance across models using Computational Accuracy (CA), Success Rate (SR), and tail-risk robustness metrics (P5-CA and $\text{CVaR}_{0.1}$). Across all evaluated models, deterministic orchestration achieves the highest CA values, indicating stronger functional equivalence to the reference implementations. These improvements are accompanied by consistently higher P5-CA and CVaR scores, demonstrating improved worst-case robustness and reduced failure severity.

Conversely, LLM-controlled orchestration yields higher SR across all models, indicating a greater likelihood of producing compilable and test-passing programs across repeated executions. However, lower P5-CA and CVaR values suggest heavier performance tails, more severe failures, and less predictable execution behavior under stochastic execution.

Taken together, these results show that orchestration strategy primarily shifts the distribution of outcomes. Deterministic orchestration reduces execution variance and improves worst-case robustness, whereas LLM-controlled orchestration increases execution success frequency while producing heavier performance tails.

### 5.3 Efficiency and Cost Analysis

The largest divergence between the two orchestration strategies is observed in computational efficiency and operational cost. While both approaches achieve comparable functional outcomes, the resource requirements for LLM-controlled orchestration are substantially higher than those of fixed execution pipelines.

*Token Consumption and Throughput.* Deterministic orchestration substantially reduces token consumption across all evaluated COBOL categories compared to the LLM-controlled baseline. As illustrated in Figure 3, LLM-controlled orchestration often requires between 1.75 million and 2.25 million tokens for complex modules such as Numeric (NC) and Sequential I/O (SQ). In contrast, deterministic orchestration completes the same tasks using approximately

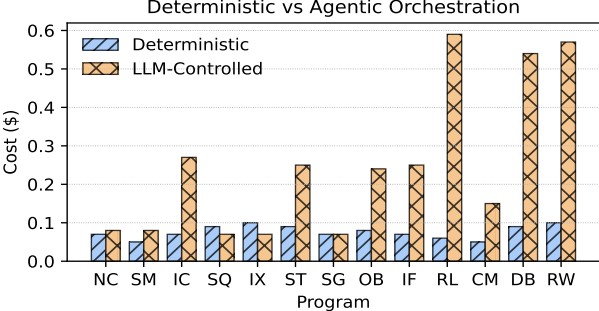

**Figure 4: Cost comparison between deterministic and LLM-controlled orchestration across COBOL benchmark programs. Each pair of bars shows the cost per successful translation for a given program under identical models, prompts, tools, and inputs, with only the execution control mechanism varied. Deterministic orchestration consistently achieves lower or comparable costs, whereas LLM-controlled orchestration incurs substantially higher costs in most cases due to increased token usage and adaptive execution paths.**

400,000 to 700,000 tokens. This represents a multi-fold reduction in token consumption for highly structured modernization tasks.

The clustering of data points near the lower y-axis in Figure 3 further suggests that deterministic control limits "agentic drift" and unbounded reasoning loops that commonly emerge in LLM-controlled systems. Small variations in model outputs can otherwise produce divergent and increasingly inefficient execution paths.

*Operational Cost Impact.* The financial implications of these efficiency gains are summarized in Figure 4, which estimates the total cost per successful translation based on token usage. The Sequential I/O (SQ) category exhibits the largest cost disparity, where LLM-controlled orchestration exceeds $140 per successful translation, while deterministic orchestration achieves the same result for approximately $40. Similar trends appear in the NC and IC categories, where agentic control significantly increases cost through repeated reasoning steps and open-ended retry behavior.

Across all modules, the "planning tax" associated with delegating workflow logic to the language model introduces a consistent cost overhead without a proportional improvement in functional correctness. These results suggest that, for enterprise-scale modernization, embedding LLMs within deterministic execution frameworks is important for maintaining economic viability and cost predictability.

### 5.4 Test-Suite Results

Table 2 summarizes results from the NIST COBOL85 test harness, comparing the original COBOL programs with Python translations generated using deterministic and LLM-controlled orchestration.

At the aggregate level, both orchestration strategies produce nearly identical execution statistics. The number of executed programs, total errors, and inspected cases remains consistent across settings, indicating comparable levels of overall functional coverage. This suggests that orchestration strategy does not materially

affect whether a translation can ultimately be produced for a given program.

Differences emerge primarily in pass and failure counts. Deterministic orchestration achieves slightly higher pass totals and fewer failures than LLM-controlled orchestration, consistent with the higher CA and robustness metrics reported in Table 1. However, these differences remain modest, reinforcing the conclusion that both strategies achieve similar baseline translation capability.

Overall, the test-suite results indicate that orchestration primarily affects consistency and robustness rather than translation feasibility. Both approaches successfully translate the same set of programs, while deterministic orchestration provides modestly stronger correctness guarantees in the final outputs.

### 5.5 Limitations

Although our experiments provide a controlled comparison between deterministic and LLM-controlled orchestration, several limitations should be considered when interpreting the results.

First, our evaluation focuses exclusively on COBOL-to-Python modernization. While this represents an important and realistic legacy migration scenario, the findings may not directly generalize to other programming languages, modernization targets, or heterogeneous enterprise systems with mixed-language dependencies. Structured legacy workflows such as COBOL modernization typically involve well-defined execution stages and validation procedures, which may amplify the benefits of deterministic orchestration relative to more exploratory software engineering tasks.

Second, the evaluation relies on the NIST COBOL85 test suite, which, although widely used and executable, does not fully capture the complexity of production enterprise environments. Real-world COBOL systems frequently interact with external infrastructure such as JCL pipelines, CICS transactions, VSAM data stores, database systems, and vendor-specific dialect extensions. These operational dependencies introduce integration challenges and environmental constraints that are not represented in standardized benchmark programs. As a result, measured correctness and robustness may overestimate performance relative to large-scale industrial modernization settings.

Third, correctness measurements depend on the completeness and quality of available test oracles. Computational Accuracy and Success Rate evaluate behavioral equivalence only with respect to the provided test cases. Programs with incomplete test coverage may appear correct despite latent semantic errors that are not exercised by the test suite. While executable benchmarks improve objectivity relative to similarity-based metrics, they do not eliminate the broader challenge of validating semantic equivalence in complex software systems.

Another limitation arises from assumptions about tooling and workflow structure. Deterministic orchestration benefits from clearly defined stages, explicit validators, and stable tool interfaces. In environments where validation mechanisms are weak, poorly specified, or unavailable, deterministic pipelines may provide fewer advantages than adaptive agentic approaches. Consequently, the observed robustness improvements should be interpreted in the context of workflows that support structured verification.

**Table 2: NIST COBOL85 test-harness summary for the original COBOL baseline and for Python translations produced under deterministic vs. LLM-controlled orchestration. Counts are aggregated by module.**

| Module | COBOL Baseline | | | | | | | | Python (Deterministic) | | | | | | | | Python (LLM-controlled) | | | | | | | |
|---|---|---|---|---|---|---|---|---|---|---|---|---|---|---|---|---|---|---|---|---|---|---|---|---|
| | Programs | Executed | Error | Pass | Fail | Deleted | Inspect | Total | Programs | Executed | Error | Pass | Fail | Deleted | Inspect | Total | Programs | Executed | Error | Pass | Fail | Deleted | Inspect | Total |
| NC | 90 | 90 | 0 | 4352 | 0 | 6 | 11 | 4369 | 90 | 90 | 0 | 4390 | 0 | 6 | 11 | 4407 | 90 | 90 | 0 | 4375 | 0 | 6 | 11 | 4392 |
| SM | 15 | 15 | 0 | 290 | 0 | 3 | 1 | 294 | 15 | 15 | 0 | 295 | 0 | 3 | 1 | 299 | 15 | 15 | 0 | 292 | 0 | 3 | 1 | 296 |
| IC | 13 | 13 | 0 | 97 | 0 | 0 | 0 | 97 | 13 | 13 | 0 | 99 | 0 | 0 | 0 | 99 | 13 | 13 | 0 | 98 | 0 | 0 | 0 | 98 |
| SQ | 81 | 81 | 0 | 512 | 0 | 6 | 81 | 599 | 81 | 81 | 0 | 525 | 0 | 6 | 81 | 612 | 81 | 81 | 0 | 520 | 0 | 6 | 81 | 607 |
| IX | 39 | 39 | 0 | 507 | 0 | 1 | 0 | 508 | 39 | 39 | 0 | 516 | 0 | 1 | 0 | 517 | 39 | 39 | 0 | 513 | 0 | 1 | 0 | 514 |
| ST | 39 | 39 | 0 | 278 | 0 | 0 | 0 | 278 | 39 | 39 | 0 | 283 | 0 | 0 | 0 | 283 | 39 | 39 | 0 | 281 | 0 | 0 | 0 | 281 |
| SG | 5 | 5 | 0 | 193 | 0 | 0 | 0 | 193 | 5 | 5 | 0 | 196 | 0 | 0 | 0 | 196 | 5 | 5 | 0 | 195 | 0 | 0 | 0 | 195 |
| OB | 5 | 5 | 0 | 16 | 0 | 0 | 0 | 16 | 5 | 5 | 0 | 17 | 0 | 0 | 0 | 17 | 5 | 5 | 0 | 16 | 0 | 0 | 0 | 16 |
| IF | 42 | 42 | 0 | 732 | 0 | 0 | 0 | 732 | 42 | 42 | 0 | 744 | 0 | 0 | 0 | 744 | 42 | 42 | 0 | 738 | 0 | 0 | 0 | 738 |
| RL | 32 | 32 | 0 | 1827 | 0 | 5 | 0 | 1832 | 32 | 32 | 0 | 1860 | 0 | 5 | 0 | 1865 | 32 | 32 | 0 | 1850 | 0 | 5 | 0 | 1855 |
| CM | 7 | 0 | 7 | 0 | 0 | 0 | 0 | 0 | 7 | 0 | 7 | 0 | 0 | 0 | 0 | 0 | 7 | 0 | 7 | 0 | 0 | 0 | 0 | 0 |
| DB | 10 | 2 | 1 | 42 | 274 | 72 | 0 | 388 | 10 | 2 | 1 | 115 | 201 | 72 | 0 | 388 | 10 | 2 | 1 | 108 | 208 | 72 | 0 | 388 |
| RW | 4 | 4 | 0 | 40 | 0 | 0 | 0 | 40 | 4 | 4 | 0 | 44 | 0 | 0 | 0 | 44 | 4 | 4 | 0 | 43 | 0 | 0 | 0 | 43 |
| Total | 382 | 367 | 8 | 8886 | 274 | 93 | 93 | 9346 | 382 | 367 | 8 | 9006 | 154 | 93 | 93 | 9346 | 382 | 367 | 8 | 8986 | 174 | 93 | 93 | 9346 |

In addition, although we control prompts, models, and decoding parameters to isolate execution control as the sole experimental variable, the results remain dependent on the selected language models and prompting configuration. Different prompting strategies, model architectures, or sampling regimes could alter the relative performance characteristics of orchestration strategies. Our conclusions therefore characterize behavior under controlled conditions rather than universal properties of all LLM-based systems.

Finally, although we provide a quantitative analysis of computational efficiency and operational cost in Figures 3 and 4, these results are primarily based on token consumption and API-based pricing. While token usage provides a consistent, model-agnostic estimate of resource requirements, it does not fully capture wall-clock latency, infrastructure overhead, or broader engineering costs associated with production deployment. In addition, our cost estimates are based on current API pricing models, which may vary across providers and over time. Future evaluations incorporating runtime measurements and longitudinal operational resource usage would provide a more complete understanding of the practical economic tradeoffs between deterministic and agentic systems.

## 5.6 Discussion

The results indicate that execution control primarily affects the distribution of outcomes rather than the overall capability of LLM-based code modernization systems. Deterministic orchestration consistently improves worst-case correctness and reduces performance variability, while LLM-controlled orchestration increases the likelihood of producing a runnable solution in a given execution. This pattern suggests that orchestration strategy governs how uncertainty introduced by stochastic language models propagates through multi-step workflows. In structured modernization tasks with fixed stages and verifiable intermediate states, constrained execution appears to limit error accumulation and produce more predictable behavior without materially reducing achievable correctness.

These findings also help clarify when agentic orchestration may or may not be advantageous. Tasks characterized by clearly defined transformation pipelines, strong validation signals, and deterministic dependencies benefit from explicit execution policies that separate execution control from code generation. In contrast, open-ended software engineering activities—such as exploratory refactoring, requirements discovery, or incomplete specifications—may still benefit from adaptive LLM-driven control, where flexibility outweighs strict reproducibility. Rather than treating deterministic and agentic approaches as competing paradigms, the results suggest that they occupy different regions of the software engineering design space.

The study supports an architectural perspective in which language models function most effectively as generative or interpretive components embedded within structured execution frameworks. Treating execution control as a stochastic reasoning problem introduces variability that may not improve correctness in workflows already governed by explicit rules and validators. This observation motivates future hybrid designs that combine deterministic default execution with carefully bounded LLM intervention, enabling adaptive behavior while preserving reproducibility, robustness, and predictable operational overhead.

## 6 Conclusion

This paper presented a controlled empirical study of deterministic and LLM-controlled orchestration for COBOL-to-Python modernization. Using the ATLAS framework, we isolated execution control as the sole experimental variable while holding language models, prompts, tools, configurations, and source programs constant. This enabled direct evaluation of how orchestration strategy affects correctness, robustness, and computational efficiency in structured software engineering workflows.

Our results show that deterministic orchestration achieves comparable functional correctness to LLM-controlled orchestration while consistently improving worst-case robustness and reducing execution variability across repeated runs. In addition, deterministic execution substantially reduces token consumption and operational cost relative to adaptive agentic workflows. These findings suggest that, in structured modernization tasks with explicit validation stages and deterministic dependencies, fixed execution policies provide more stable and cost-efficient behavior without sacrificing translation quality.

The results indicate that execution control should be treated as a first-class systems design decision rather than an implicit consequence of language model reasoning. While adaptive agentic orchestration remains valuable for open-ended and exploratory tasks, structured modernization workflows benefit from separating

execution control from generative reasoning. This distinction has important implications for the design of reliable, scalable, and economically sustainable LLM-based software engineering systems.

Future work will extend this analysis to additional programming languages, heterogeneous enterprise environments, and hybrid orchestration architectures that combine deterministic execution with bounded adaptive reasoning. Further investigation of long-horizon execution behavior, runtime efficiency, and production-scale deployment characteristics will also provide deeper insight into the tradeoffs between deterministic and agentic software engineering systems.

## Acknowledgments

We would like to thank Alan Marchiori, Todd Schmid, and Hung Pham for their valuable feedback.

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
