# OpenReview forum: "Deterministic vs. LLM-Controlled Orchestration for COBOL-to-Python Modernization"
_ACM.org/AIWare/2026/Conference — AIware 2026_

### Official Review · Reviewer_HEbb · 2026-03-07

**Rating:** 4
**Confidence:** 4

**Review:**

## Pros

* The paper studies an important architectural design question in LLM-based software engineering systems, namely, how execution control should be handled in multi-step tool pipelines.

* The experimental setup attempts to isolate the orchestration strategy as the only variable, which helps reduce confounding factors in the comparison.

* The evaluation uses an executable benchmark with a test harness, allowing correctness to be measured through program execution rather than heuristic similarity metrics.

* The paper analyzes not only correctness but also robustness and computational cost, providing a broader perspective on system behavior.


## Cons

* The technical novelty is limited because the work mainly compares two orchestration strategies rather than introducing a new algorithm or system design.

* The study focuses exclusively on COBOL-to-Python modernization using the NIST COBOL85 benchmark, which may not reflect the complexity of real enterprise modernization scenarios.

* The reported differences in correctness between the two orchestration strategies are relatively small, raising questions about the practical significance of the findings.

* The evaluation relies on a single benchmark suite and does not include other code translation tasks or modern software engineering workflows.

---


## Quality and Soundness

The experimental setup attempts to control most variables so that execution control becomes the only difference between the two approaches. However, the evaluation is limited to a single benchmark dataset and a specific modernization task. The NIST COBOL85 test suite provides executable validation, but it does not represent the complexity of real production systems with external dependencies, infrastructure integration, or incomplete specifications. As a result, the reported improvements in robustness and efficiency may not generalize to broader software engineering tasks.

## Clarity

The paper is generally understandable, and the architecture of the ATLAS system is illustrated with diagrams that describe the deterministic and agentic orchestration pipelines. However, some design decisions are described at a conceptual level without sufficient implementation detail. For example, the exact mechanisms used by the LLM-controlled orchestration to select tools and determine execution order are not fully specified. Providing clearer examples of execution traces or orchestration decisions would improve transparency.

## Originality

The originality of the work is limited. The paper mainly performs an empirical comparison between deterministic pipelines and LLM-driven orchestration strategies. While the controlled experiment is useful, the study does not introduce new orchestration algorithms, planning mechanisms, or system architectures. The contribution is therefore primarily observational rather than methodological.

## Significance

The study highlights an important tradeoff between deterministic execution pipelines and agentic tool orchestration in LLM-based development systems. However, the practical implications are somewhat constrained by the narrow evaluation setting. The experiments focus on a single modernization task and do not demonstrate whether similar behavior occurs in other types of LLM-based software engineering workflows, such as debugging, refactoring, or automated development pipelines. As a result, the broader impact of the findings remains uncertain.

**Summary:**

This paper studies the impact of execution control strategies in LLM-based code modernization systems. The authors argue that many recent systems adopt agentic designs where the language model dynamically orchestrates tool usage, while traditional software engineering workflows rely on deterministic execution pipelines. To investigate this difference, the paper implements two orchestration strategies within the ATLAS framework: deterministic tool orchestration with a fixed execution pipeline, and LLM-controlled orchestration where the model selects tools and execution steps dynamically.

The study focuses on COBOL-to-Python code modernization and evaluates both approaches using the NIST COBOL85 test suite. The experiments hold the language model, prompts, tools, and inputs constant so that execution control is the only experimental variable. The evaluation measures functional correctness using Computational Accuracy and Success Rate, robustness through tail-risk metrics, and efficiency through token consumption and estimated cost.

The results show that deterministic orchestration achieves slightly higher computational accuracy and stronger worst-case robustness, while LLM-controlled orchestration achieves higher success rates across repeated runs. Deterministic orchestration also significantly reduces token consumption and estimated operational cost. :contentReference[oaicite:0]{index=0}

---

> ### Author Response · Authors · 2026-03-21
>
> We thank the reviewer for the balanced and detailed evaluation.
>
> Limited novelty: We agree that the contribution is not algorithmic. The main contribution is a controlled empirical study that isolates execution control and examines its effect on robustness, variability, and cost. This will be stated more clearly.
>
> Limited evaluation scope: We agree. The focus on a controlled setting is intentional. This limitation will be made more explicit.
>
> Small correctness differences: We agree that the difference in mean correctness is small. The main effect is in variability where deterministic orchestration reduces variance and improves worst-case behavior, while agentic orchestration improves success rate but shows higher variability. This will be emphasized.
>
> Clarity of agentic mechanism: We agree that more detail will help. In the agentic setting, tool selection and execution order depend on model outputs, leading to variation across runs. This will be illustrated with examples and detailed in our GitHub repo.
>
> Broader impact: We will expand the discussion to distinguish between structured workflows, where deterministic execution works well, and exploratory workflows, where adaptive approaches are more suitable.

---

### Official Review · Reviewer_L3dC · 2026-03-10

**Rating:** 3
**Confidence:** 4

**Review:**

## Strengths

+ Practical Relevance: The paper tackles a highly critical and timely software engineering problem: the modernization of legacy codebases (COBOL). Providing empirical evidence on how to build more reliable and cost-effective LLM pipelines for this task is of great value to the industry.

+ Rigorous Experimental Design: The methodology is commendable. By carefully isolating the "orchestration strategy" from other confounding variables, the authors ensure a fair and scientifically rigorous comparison.

+ Actionable Insights: The findings regarding token efficiency and execution robustness provide clear, actionable guidance for researchers and practitioners building LLM-based software engineering tools.

## Weaknesses

- Mismatch between motivation and evaluation benchmark complexity. The paper motivates the research by highlighting the massive scale and complexity of real-world legacy systems. However, the evaluation is exclusively conducted on the NIST COBOL85 test suite. These benchmark programs are typically self-contained and focus on language features or algorithms. They fail to represent the true complexity of enterprise-grade COBOL systems, which are heavily characterized by massive file dependencies, complex database interactions, and coupling with other legacy frameworks. Consequently, the claim that deterministic orchestration is superior for "legacy system modernization" feels slightly overstated.

- Potential under-representation of Agentic capabilities. In the pursuit of a strictly controlled experiment, the authors constrained the agentic system to use the exact same prompts, context, and toolsets as the deterministic pipeline. While this ensures fairness on paper, it may artificially handicap the agentic approach. The true strength of an agentic system lies in its ability to dynamically leverage specialized tools (e.g., semantic code search, syntax checkers, dynamic test generators) and utilize complex self-reflection/reasoning prompts. If provided with a richer set of actionable tools and a broader context window tailored for autonomous debugging, the agentic baseline might have performed significantly better, especially in edge cases where the rigid deterministic pipeline fails.

## Questions and Suggestions

+ Given that real-world COBOL projects involve multiple inter-dependent files, how would your deterministic orchestration scale? Have you tested cases where the translation requires cross-file context retrieval?

+ Have you considered evaluating a "hybrid" orchestration approach? For instance, using a deterministic pipeline for the macro-level translation workflow, but delegating specific micro-tasks (such as resolving a stubborn syntax error after a failed test) to an agentic loop. Discussing this could strengthen the paper's forward-looking perspective.

+ Suggestion: I recommend acknowledging the limitations of the NIST COBOL85 dataset in the discussion section and explicitly scoping the claims to self-contained code translation, rather than broad enterprise modernization.

**Summary:**

This paper presents a well-controlled empirical study comparing two different tool orchestration strategies for the task of LLM-based code translation, specifically focusing on modernizing COBOL to Python. By strictly controlling variables such as the underlying LLMs, prompts, toolsets, and inputs, the authors isolate the impact of the orchestration method itself. The results indicate that while both approaches achieve comparable baseline translation correctness, the deterministic orchestration significantly improves worst-case robustness, reduces performance variability, and drastically cuts down computational costs.

---

> ### Author Response · Authors · 2026-03-21
>
> We thank the reviewer for the insightful comments.
>
> Benchmark vs. enterprise complexity: We agree that COBOL85 does not represent full enterprise complexity. It is used here to allow controlled evaluation with executable correctness checks. We will clearly state that the conclusions are limited to structured, self-contained translation tasks.
>
> Agentic baseline constraints: The use of the same models, prompts, tools, and inputs is intentional to isolate execution control. Allowing additional tools or prompt changes would introduce confounding factors. We will clarify and suggest this as future work.
>
> Multi-file scaling: This is an important direction. While not part of the current study, the approach can be extended to multi-file settings that handle dependencies and retrieval. This will be included in our plan for future work.
>
> Hybrid orchestration: We agree and will include a short discussion of hybrid approaches, linking them to the observed trade-offs.
>
> Scope of claims: We agree and will further tighten the scope of the claims.

---

### Official Review · Reviewer_rvSG · 2026-03-13

**Rating:** 3
**Confidence:** 4

**Review:**

Strengths:

The controlled experimental design. The authors hold models, prompts, tools, and inputs constant and vary only the execution control mechanism, enabling a clean causal analysis of orchestration effects.

The paper is clear and easy to follow.

The work addresses a timely and important question: whether agentic LLM orchestration actually improves software engineering workflows. Many recent systems assume that LLM agents should control execution, yet this assumption is rarely empirically tested.
The use of repeated runs with stochastic seeds to analyze robustness is particularly valuable because it captures variability introduced by LLM behavior.

Providing practical suggestions for enterprise-scale setup adds to the value f the paper.

Limitations:

Strengthen the research contribution by formalizing a taxonomy of orchestration strategies or a hybrid orchestration model combining deterministic control with bounded agent reasoning.

While this is a realistic modernization task, the conclusions may not generalize to exploratory tasks (debugging, refactoring), incomplete specifications, multi-repository development workflows.

The paper shows that agentic orchestration increases cost but does not deeply analyze why agentic reasoning produces longer execution traces. Potential missing analyses include tool-call sequence analysis, failure mode clustering or reasoning-step breakdown.

The implementation of the ATLAS system is not provided in the paper. Since the contribution relies heavily on experimental comparison between orchestration strategies, I strongly suggest releasing the implementation or an artifact (e.g., repository or replication package) to improve reproducibility and transparency.

In addition, the paper states that the same prompts are used across different models; however, the actual prompts, system instructions, and assistant role configurations are not disclosed.

Finally, to strengthen the paper’s contribution, I suggest presenting the proposed LLM design guidelines in a more structured manner. For example, the guidelines could be explicitly enumerated and then connected to the taxonomy , as suggested in one of my earlier comments.

**Summary:**

The paper studies the impact of execution control strategies in LLM-based software engineering systems, specifically comparing deterministic tool orchestration with LLM-controlled (agentic) orchestration for the task of COBOL-to-Python legacy code modernization. The authors implement both approaches within the same framework (ATLAS) and conduct controlled experiments where the only varying factor is the orchestration mechanism. The evaluation uses the NIST COBOL85 test suite, measuring correctness (Computational Accuracy and Success Rate), robustness across repeated runs, and efficiency (token usage and cost). The results show that deterministic orchestration achieves comparable correctness but improves worst-case robustness and significantly reduces token consumption and operational cost. The paper concludes that structured software engineering workflows benefit from deterministic execution policies rather than fully agentic orchestration

---

> ### Author Response · Authors · 2026-03-21
>
> We thank the reviewer for the helpful feedback and for recognizing the study's clarity.
>
> Taxonomy/hybrid orchestration: The current study focuses on two endpoints to clearly isolate execution control. At the same time, the results naturally point to a broader design space. Deterministic execution provides stability and bounded behavior, while agentic control helps in recovery. This suggests a hybrid setting in which execution remains structured, but limited agent reasoning is used when needed. This perspective will be made clearer in the revised version.
>
> Generalization beyond structured tasks: We agree with this point. The results apply to structured workflows with fixed stages and verifiable intermediate outputs. This may not hold for exploratory settings such as debugging or open-ended development. This distinction will be stated more clearly.
>
> Analysis of agentic cost: We agree that the explanation can be improved. The cost difference comes from how execution proceeds. In the agentic setting, execution steps are generated dynamically, leading to longer, sometimes repetitive, tool use. In contrast, deterministic execution follows a fixed sequence and avoids such repetition.
>
> Implementation and reproducibility: The implementation is complete, but was not included due to double-blind requirements. The code, prompts, and evaluation setup will be shared via a GitHub repository for reproducibility.
>
> Prompts and system instructions: Both approaches use the same prompts and configurations to ensure a fair comparison. Representative examples will be included for clarity. Complete details will be shared via GitHub repository.
>
> Structured design guidelines: The design insights are currently described in text. These will be reorganized into clear and structured guidelines.

---

### Author Response · Authors · 2026-03-21

We thank the reviewers for their careful reading and constructive feedback. We are encouraged by the positive assessment of the controlled experimental design, executable evaluation, and the analysis of robustness and cost. The goal of this work is to study execution control as a design variable and understand how it affects correctness, variability, and cost under controlled conditions. Below, we respond to the comments and clarify the scope and intended revisions.